# Synthesis and Characterization of Silver–Strontium (Ag-Sr)-Doped Mesoporous Bioactive Glass Nanoparticles

**DOI:** 10.3390/gels7020034

**Published:** 2021-03-24

**Authors:** Shaher Bano, Memoona Akhtar, Muhammad Yasir, Muhammad Salman Maqbool, Akbar Niaz, Abdul Wadood, Muhammad Atiq Ur Rehman

**Affiliations:** 1Department of Materials Science and Engineering, Institute of Space Technology Islamabad, Islamabad 44000, Pakistan; sbanozaidi@yahoo.com (S.B.); memoonabajwa94@gmail.com (M.A.); muhammadyasir85@gmail.com (M.Y.); wadood91@gmail.com (A.W.); 2Department of Mechanical and Manufacturing Engineering, La Trobe University, Melbourne, VIC 3086, Australia; salman.maqbool87@gmail.com; 3Department of Mechanical Engineering, King Faisal University, Al Hufūf 31982, Saudi Arabia; abutt@kfu.edu.sa

**Keywords:** mesoporous bioactive glass nanoparticles, sol-gel, antibacterial activity, silver, bioactivity

## Abstract

Biomedical implants are the need of this era due to the increase in number of accidents and follow-up surgeries. Different types of bone diseases such as osteoarthritis, osteomalacia, bone cancer, etc., are increasing globally. Mesoporous bioactive glass nanoparticles (MBGNs) are used in biomedical devices due to their osteointegration and bioactive properties. In this study, silver (Ag)- and strontium (Sr)-doped mesoporous bioactive glass nanoparticles (Ag-Sr MBGNs) were prepared by a modified Stöber process. In this method, Ag^+^ and Sr^2+^ were co-substituted in pure MBGNs to harvest the antibacterial properties of Ag ions, as well as pro-osteogenic potential of Sr^2^ ions. The effect of the two-ion concentration on morphology, surface charge, composition, antibacterial ability, and in-vitro bioactivity was studied. Scanning electron microscopy (SEM), X-Ray diffraction (XRD), and Fourier transform infrared spectroscopy (FTIR) confirmed the doping of Sr and Ag in MBGNs. SEM and EDX analysis confirmed the spherical morphology and typical composition of MBGNs, respectively. The Ag-Sr MBGNs showed a strong antibacterial effect against *Staphylococcus carnosus* and *Escherichia coli* bacteria determined via turbidity and disc diffusion method. Moreover, the synthesized Ag-Sr MBGNs develop apatite-like crystals upon immersion in simulated body fluid (SBF), which suggested that the addition of Sr improved in vitro bioactivity. The Ag-Sr MBGNs synthesized in this study can be used for the preparation of scaffolds or as a filler material in the composite coatings for bone tissue engineering.

## 1. Introduction

Millions of medical devices are being implanted nowadays in patients related to bone diseases and accidental surgeries, thanks to the advancement of biomaterials. During the last few decades, biomaterials have focused on the following issues: (a) establish a material with a suitable mechanical strength, and (b) improve in-vitro activity by increasing the surface area of the bio-ceramics [1,2]. The number of accidents worldwide are increasing periodically. Moreover, the percentage of people over 50 years that are suffering from diseases like osteoporosis, osteoarthritis, osteomalacia, bone cancer, and other musculoskeletal diseases has increased [3]. It is stated that only in the USA, annually, more than 500,000 primary arthroplasties including total joint replacement, total hip arthroplasty (THA), and total knee arthroplasty (TKA) are done, and more than 1.3 million people live with artificial joints [4].

Bioactive glasses (BGs), new generation of bio-ceramics, are preferred biomaterials in a wide range of biomedical applications such as the regeneration of hard tissues (bones and drug delivery), owing to their bioactivity (in vitro and in vivo), osteoconductivity, osteoinductivity, high degradation rate, tailorable morphology (e.g., miniature size, high specific surface area, pore structure), and favorable physicochemical properties (e.g., bone bonding potential, biodegradability) [5,6,7]. BGs can resorb in physiological fluids and their degradation products stimulate the osteogenesis through increasing cell proliferation and expression of osteogenic marker genes. Composition, pore size, particle size, and the specific surface area are the main factors on which the bioactivity and degradation rate strongly depends. These required characteristics can be obtained by employing mesoporous bioactive glass nanoparticles (MBGNs).

MBGNs are produced commonly either via the conventional melt quench method or by using the sol-gel method [8]. The melt quench method facilitates scalability in MBGNs production. However, it produces inhomogeneous particles in size and shape. Moreover, sophisticated equipment is required for the melt quench approach. These issues discourage the wide adoption of this technique for MBGNs production. On the other hand, the sol-gel method enables to produce MBGNs at relatively low temperature with controlled shape and size without using any complicated equipment [9].

The physicochemical properties (such as mechanical properties, apatite-forming capability) of MBGNs can be improved by incorporating peculiar metallic elements [10,11,12]. In the last decades, inclusion of metallic ions like aluminum (Al), zirconium (Zr), magnesium (Mg), strontium (Sr), lithium (Li), zinc (Zn), and silver (Ag), are being used as a substitution of Ca in MBGNs at low dosage, and can cause promoted functionalities [13,14]. For instance, the addition of Ag improves the antibacterial properties of bioactive glass nanoparticles (BGNs), whilst the existence of Li improves the osteogenic activities. Numerous metallic ions like copper, gallium, strontium, cobalt, cerium, and zinc assimilated in bioactive glasses to boost bone formation because of their effect on osteogenesis and angiogenesis.

Strontium is an essential trace element in the body, which accumulates frequently in bones, with a fraction of about 0.035% of the whole calcium. Strontium showed a twofold job in the bone remodeling process by ever-increasing bone formation and lessening bone resorption [15,16].

Ag is incorporated in BGs to prepare implants and scaffolds for tissue engineering owing to its exceptional antibacterial property. Ag ions can stick with the cell wall and cytoplasmic membranes due to strong affinity with sulfur proteins. These stick Ag ions may enhance the permeability of cytoplasmic membranes, which in turn causes disruption of bacterial envelope [17]. Monodispersed BG microspheres tailored with silver nitrate (AgNO_3_) are synthesized by a modified Stöber’s method, which shows antibacterial performance [18]. MBGNs assimilated with Ag are supportive for ion release and enhanced bacterial inhibition. Ag-doped BG is a promising dental material as dental implants are frequently exposed to bacteria [19,20,21]. MBGNs combined with Ag nanoparticles indicate a bactericidal effect against *Enterococcus faecalis*, which exists in the root canal system without inducing cytotoxicity on mesenchymal stem cells (MSCs). Therefore, Ag-doped MBGNs are potential building blocks for preparing orthopedic implants. Bari et al. studied MBGNs with admirable textural properties, in vitro bioactivity, and excellent antibacterial properties against different bacteria [20,22,23,24,25,26].

The research work presented in this study involves the synthesis and characterization of novel Ag- and Sr-containing mesoporous bioactive glass nanoparticles (Ag-Sr-MBGNs) via a modified Stöber process [27]. MBGNs were initially doped with various concentrations of Sr-Ag (5:1) mole%. Preliminary in vitro bioactivity studies confirmed the suitability of Ag-Sr-MBGNs for bone tissue engineering applications, through formation of hydroxyapatite crystals upon immersion in simulated body fluid (SBF). The controlled release of Ag-Sr ions also induced the antibacterial characteristics without affecting the bioactivity of MBGNs. The antibacterial effect correlates with the release of metallic ions in a critical concentration of ions (Ag), which works against the relevant pathogen or bacteria in physiological conditions. Therefore, the results presented in this article are anticipated to be used for a way forward in development of third-generation biomaterials with the application of Ag-Sr-based MBGNs for scaffold fabrication as well as antibacterial coatings on metallic substrates. Osteogenic properties of other ions like Cu, Mn, and Zn, as well as their antibacterial studies, have also been studied and their cytotoxic effects are also highlighted in the literature [20].

## 2. Results and Discussion

### 2.1. Morphological Characterization

The morphology of the synthesized MBGNs was investigated by SEM analysis. Figure 1 shows that all types of MBGNs have spherical morphology regardless of the addition of metallic precursor. Figure 1 depicts that average particle size of synthesized MBGNs was 130 ± 15 nm. The microemulsion-assisted sol-gel method favors the dispersion of nanoparticles, which explains the homogeneous size and shape of the obtained nanoparticles [28].

Figure 2 (left) confirms the mesoporous nature of the synthesized MBGNs (with different concentrations). Figure 2 (right) depicts the nitrogen adsorption and desorption isotherm of Sr-MBGNs, Ag MBGNs, and Ag-Sr MBGNs. Textural properties of Ag, Sr, and Ag-Sr MBGNs derived from nitrogen adsorption-desorption isotherm analysis depicts a type IV isotherm according to IUPAC (International Union of Pure and Applied Chemistry) which confirms the mesoporous structure [26]. Uptake of a high amount of nitrogen at relative pressure (P/Po) ≈ 0.99 indicates the nano-sized particles. It was deduced from Figure 2 that all the particles exhibit wide pore size range with the average pore size of ~2.8 nm. Relatively high porosity in the synthesized MBGNs may lead to the high surface area [29]. This porous nature opens up other biomedical applications such as drug delivery and microbial cell encapsulation [25].

### 2.2. Compositional Analysis

The EDX analysis was conducted to confirm the addition of Ag and Sr in MBGNs. For EDX analysis, powder samples were used. MBGNs powder was dispersed in ethanol and then ultra-sonicated for half an hour in order to avoid agglomerates. After drying, EDX analysis was conducted.

Figure 3A represents the peaks of Ag and Sr, which confirmed the substitution of Ag and Sr in MBGNs. Figure 3B represents the qualitative elemental EDX analysis of the synthesized MBGNs prior to the substitution of Ag and Sr ions. It was observed that the Ca and Si peaks are present in MBGNs, which indicated the formation MBGNs [30]. The results of EDX analysis are in good qualitative agreement with the nominal composition of the synthesized MBGNs.

The molecular structure of as-synthesized Ag-Sr MBGNs and the effect of doping on a network of glass were studied by FTIR spectroscopy, as shown in Figure 4. The results depicted that no major difference occurs upon doping metallic precursors [30]. The bands around 455 and 1067 cm^−1^ can be assigned to Si–O–Si rocking and Si–O–Si stretching modes, respectively [31]. The broad band at 1200 to 1000 cm^−1^ depicts Si–O–Si vibrations [32]. The peak around 800 cm^−1^ is assigned to the Si–O–Si bridging bonds in the SiO_4_ tetrahedrons [33].

The XRD diffraction pattern of as-synthesized MBGNs, Ag MBGNs, Sr MBGNs, and Ag-Sr MBGNs confirmed the amorphous nature (broad peak at 2θ = 20°–32°) for all types of MBGNs, as shown in Figure 5 [34]. Furthermore, the diffraction pattern of Ag-Sr MBGNs shows no peaks ascribed to the silver and strontium, which suggests the incorporation of Ag and Sr into MBGNs as well as the chemical homogeneity of Ag-Sr-containing MBGNs. It was concluded that Ag-Sr MBGNs were successfully synthesized using the microemulsion-assisted sol-gel approach presented here, with silver nitrate and strontium nitrate being effective precursors for incorporating Ag and Sr into the silica network of MBGNs.

### 2.3. Zeta Potential

The zeta potential measurements of MBGNs, Ag MBGNs, Sr MBGNs, and Ag-Sr MBGNs were performed in ethanol, and the results are given in Table 1. It was deduced that silver and strontium ions changed the surface charge of MBGNs. Strontium substitution resulted in an increase in positive surface charge, while Ag substitution led to a decrease in surface charge. The variation in zeta potential by the incorporation of Sr and Ag in MBGNs may be associated with the pH change (zeta potential is a function of pH). It is also reported that the addition of Sr in bioactive glass (Sr substitution with Ca) may lead to the pH change and eventually increase the zeta potential compared to the MBGNs [35]. Positive zeta potential (of Sr-MBGNs) increases the solubility of the nanoparticles and may lead to aggregation. However, it also promotes the adsorption of negatively charged proteins on the surface and improves the efficacy of imaging, gene transfer, and drug delivery [36]. Ag ion reduces the surface charge of MBGNs due to its relatively high electronegativity (1.93) compared to calcium (1.0), which facilitates the deposition of Ca^2+^ ions on the surface and enhances the bioactivity [37].

### 2.4. Ion-Release Profile

The synthesized MBGNs were tracked for ion-release study in order to understand the effect of ion release on the biological properties, for example, antibacterial activity, in vitro bioactivity, and cell biology. Figure 6A represents the release of Si and Ca ions from MBGNs. It was observed that Si showed a rapid release in all samples in the first 7 days, followed by a relatively slow release up to 21 days. Ca^2+^ ions were released at a rapid rate from all types of MBGNs. However, the absolute release of Ca ions decreases with the increase in the incubation time. The release of Ca ions is beneficial for the osteoconductive properties of the bioactive glasses.

Figure 6B shows the release profile of Ag and Sr ions from co-substituted MBGNs under dynamic condition in SBF solution at 37 °C over a period of 21 days. A burst release of Ag ions was observed in the first 24 h in Ag-Sr MBGNs and Ag MBGNs samples followed by a steady-state release, indicating long-term sustained release, which will be beneficial for long-term antibacterial effect. Figure 6C shows the release profile of Si, Ca, and Ag ions from the Ag MBGNs. We observed a burst release of Ag ions during the first week. Afterwards, the sustained release of Ag ions was observed. The ion release profile of Si, Ca, and Ag from Ag MBGNs was similar to the Ag-Sr MBGNs. Furthermore, the release of Sr, Si, and Ca from Sr MBGNs was similar to that of Ag-Sr MBGNs. Thus, it was concluded that the co-substitution of Ag and Sr did not affect the release of Si and Ca ions, which will be helpful in obtaining bioactive properties while keeping the antibacterial effect associated with the release of Ag ions.

The burst release of both ions (Ag, Sr) was observed from Ag-Sr MBGNs, which might be due to the concentration gradient between the particles and physiological solution. Ag ions released in the physiological medium play an important role in the antibacterial activity. The antibacterial properties (discussed in Section 2.5) of the synthesized MBGNs were in good agreement with the ion-release data. Ag ions released from Ag-Sr MBGNs samples were within the concentration range of 2–48 ppm, which has been proven to induce significant antibacterial properties against Gram-positive and Gram-negative bacterial strains [38]. The sustained release of Sr ions will be beneficial for in vitro bioactivity [39]. Furthermore, the initial burst release of silver will be useful in preventing the formation of biofilm and the sustained release of Ag will be effective in providing a long-term antibacterial effect. It was observed that after 21 days of incubation, the silver release was in the range of minimum inhibitory concentration level [10]. Furthermore, in the future, it would be interesting to analyze the release of P ions because the consumption of phosphate ions from SBF confirms the Hydroxyapatite (HA) formation.

### 2.5. Antibacterial Study (Turbidity Test)

To investigate the antimicrobial effect of synthesized nanoparticles of different compositions, a turbidity test was done. The change in OD_600_ after 1, 2, 3, 4, 6, and 24 h of incubation is presented in Table 2. It was observed that the measured OD_600_ value for the Ag-Sr MBGNs and Ag MBGNs showed a strong decrease after 6 h of incubation compared to the control samples (MBGNs and Sr MBGNs). Since Ag ions released a substantial amount after 6 h of incubation, which resists the growth of *E. Coli* and *S. carnosus*, after 24 h of incubation, the cumulative release of Ag ions from Ag MBGNs and Ag-Sr MBGNs was sufficient to completely hinder the growth of *E. Coli* and *S. carnosus*. Moreover, it was observed that the control samples allowed the growth of *E. Coli* and *S. carnsus* after 24 h of incubation. Thus, it can be concluded that the Ag-Sr MBGNs and Ag MBGNs strongly retarded the growth of E. Coli cells [40].

### 2.6. Disc Diffusion Test (Inhibition Halo Method)

The antibacterial properties of the Ag-Sr MBGNs and MBGNs were also investigated by the disc diffusion method (Figure 7) to further validate the antibacterial results. The antibacterial effect was tracked against Gram-negative (*E. coli*) and Gram-positive (*S. carnosus*) bacteria. The growth of *E. coli* and *S. carnosus* was prominent on the reference and pure MBGNs sample after 24 h of incubation. The growth for both types of bacteria was strongly inhibited by the Ag-Sr MBGNs samples. Figure 7 shows that the zone of inhibition developed across the MBGNs sample against *S. carnosus* and *E. coli*. The strong antibacterial effect associated with the Ag-Sr MBGNs was due to the release of Ag ions (as shown in Figure 6). The Ag ions interact with nucleic acids and they interact preferentially with the bases in DNA, thus inhibiting the DNA replication activity and eventually leading to the death of bacteria. Furthermore, Ag in an ionic form is highly reactive (generation of reactive oxygen species) and can rupture the walls of bacteria and lead to the death of bacteria cells [1,11].

In the current study, Ag was successfully doped in the network of MBGNs (Figure 6) and the Ag was released in an ionic form rather than the particulate form. Ag in the form of particles is toxic to the osteoblast cells. However, the controlled release of silver ions <100 ppm (as the case in the present study, see Figure 6) provided a potent antibacterial effect against a wide spectrum of bacteria. The release of silver ions was <100 ppm, which is below the cytotoxic limit of Ag [10,35].

The cytotoxicity of Ag-doped MBGNs depends on the concentration of Ag in MBGNs and the release profile of Ag [33,41]. However, the cytotoxic effect associated with the release of Ag ions can be co-doped by the addition of Sr ions. In our previous study, we have shown that the toxic effect of Ag can be minimized by the co-substitution of Sr and Mn along with the Ag [1,6,21]. Therefore, this study presents a new frontier in the field of biomedical materials by the use of co-substituted Ag and Sr ions. The co-substitution of Ag and Sr is a challenging task because Ag tends to oxidize readily and form AgO. However, in this study, we developed MBGNs doped with Ag in its pure form (XRD results indicate no crystalline peak of silver oxide), due to which it was possible to release silver in an ionic form rather than particulate form [10].

### 2.7. In Vitro Bioactivity Analysis

Bioactivity is one of the most desired attributes for bone tissue engineering (BTE). The ability of the coating to form a bond with the bone is crucial for an implant [42]. Figure 8 represents the EDX analysis of the Ag-Sr MBGNs after immersion in SBF (simulated body fluid, by Kokobu et al. [42]). The decrease in the intensity of Si peak over the immersion time in SBF may indicate the degradation of Ag-Sr MBGNs or the formation of a thick layer of hydroxyapatite (HA) [5]. Moreover, it was observed that the intensity of calcium and phosphate peaks increased over the incubation time, which indicates the formation of HA crystals on the surface of the Ag-Sr MBGNs [43]. In vitro bioactivity of pure MBGNs is illustrated in our previous studies [26]. Furthermore, the toxic effect of Ag MBGNs on the bioactivity was also illustrated in our previous studies [1,6].

Figure 9 shows the SEM images of the synthesized Ag-Sr MBGNs after immersion in SBF. Figure 9 depicts the change in the morphology of nanoparticles. After 7 days of immersion in SBF, nanostructure and porous HA crystals formed on the surface of the particles. It was further observed that the plate-like HA crystals form on the surface of incubated HA. The plate-like structure indicated the calcium-enriched apatite crystals [12]. The FTIR analysis of Ag-Sr MBGNs after immersion in SBF was not investigated in the current study. However, in our recent study, we presented the FTIR analysis of Ag-Sr MBGNs incorporated in the chitosan/gelatin matrix after immersion in SBF. It was observed that the carbonate- and phosphate-related peaks appear after immersion in SBF [1].

Studies show that bone reformation is pH-sensitive. During bone remodeling around the border of osteoclast, pH is 4.0 and the pH of surrounding body fluid is 7.4 [44,45]. Moreover, it is known that the physiological environment of initial fracture hematoma is acidic, and during healing, it becomes alkaline, which aids bone differentiation [46]. To study the pH changes, Ag-Sr MBGNs samples were immersed in SBF and then incubated. SBF solution was changed after every 3 h. Initially, pH of the SBF was set at 7.40 ± 0.02, and later, the pH was checked at the 3rd, 7th, 14th, and 21st days. Figure 10 shows that the pH became slightly basic as the immersion time increased, which aids bone differentiation, as mentioned earlier. The overall curve progression is stable except for a few midterm fluctuations.

On the basis of in vitro bioactivity and antibacterial studies, it was inferred that the Ag-Sr MBGNs provided a potent antibacterial effect while maintaining the typical bioactivity associated with the MBGNs. According to Reference [41], Ag may affect the in vitro bioactivity of the MBGNs. Thus, the addition of Sr improved the in vitro bioactivity and provided an antibacterial effect due to the Ag doping.

## 3. Conclusions

In this study, we synthesized Ag-doped, Sr-doped, and Ag-Sr-doped MBGNs via modified Stöber method and sol-gel process. SEM images confirmed the spherical morphology of all the synthesized particles. BET results confirmed the mesoporous nature of all the synthesized MBGNs. It was deduced that the addition of metallic ions did not affect the morphology of MBGNs. Furthermore, XRD results confirmed the doping of Ag and Sr in the silica network of MBGNs. The XRD patterns confirmed the amorphous nature of the synthesized MBGNs with all the different concentrations. The release of Ag and Sr ions was tracked by the ICP studies. The results confirmed that during the first day of incubation, Ag and Sr showed a burst release. However, with the increase in incubation time, Ag and Sr were released in a sustained manner, thus providing a long-term therapeutic effect. The controlled release of Ag provided a potent antibacterial effect, while the release of Sr ions improved the in vitro bioactivity. The peculiar morphological features of the synthesized Ag-Sr MBGNs and the feasibility of functionalizing these MBGNs with active ions or biomolecules suggest that the synthesized MBGNs based on SiO_2_-CaO in this study are a promising material for biomedical applications, including bone regeneration and wound cure.

## 4. Materials and Methods

### 4.1. Materials

Tetraethyl orthosilicate (TEOS) 99% (Sigma Aldrich, Steinheim, Germany), calcium nitrate (Ca (NO_3_)_2_.4H_2_O) 98% (Sigma Aldrich, Steinheim, Germany), silver nitrate (Ag (NO_3_)_2_) 99% (Sigma Aldrich, Steinheim, Germany), and strontium nitrate (Sr (NO_3_)_2_) 99% (Sigma Aldrich, Steinheim, Germany), were used as silicon, calcium, silver, and strontium sources, respectively. Furthermore, ethyl acetate 99.8% (Sigma Aldrich, Steinheim Germany), cetyltrimethylammonium bromide (CTAB) 98% (Merck, Billerica, MA, USA), ammonium hydroxide 35% (VWR, Shanghai, China), distilled water, and absolute ethanol 99.8% were used. All chemicals used were of analytical grade.

### 4.2. Synthesis of Ag-Sr-Containing MBGNs (Stöber Process)

Ag-Sr MBGNs were prepared by a modified Stöber process [33,47]. Firstly, 0.56 g CTAB was dissolved in 26 mL of distilled water under continuous stirring for 30 min at 40 °C. Then, 8 mL ethyl acetate was added dropwise into the solution. Thirdly, 26 mL of diluted solution of ammonium hydroxide (32 vol.%) was added to maintain pH at 9.5 and 6 mL of TEOS and was added dropwise into the above solution under continuous stirring. Then, 2.24 g calcium nitrate, 0.0834 g silver nitrate, and 0.42 g strontium nitrate was added depending upon the required composition and followed by magnetic stirring for 30 min. Afterwards, the solution was allowed to pursue reaction between the reactants. Subsequently, the suspension was centrifuged at 7830 rpm for 10 min to separate particles from the parent solution, followed by washing the sedimented particles with ethanol. This step was repeated three times. Finally, the precipitates were dried in an oven at 75 °C for 12 h, followed by calcination at 700 °C for 5 h. Figure 11 illustrates the synthesis of Ag-Sr-doped MBGNs.

In this study, MBGNs with three different compositions were synthesized, i.e., MBGNs doped with 5 mol% Sr (5Sr-MBGNs), 1 mol.% Ag (1Ag-MBGNs), and 5 mol.% Sr and 1 mol.% Ag (5Sr-1Ag MBGNs). Table 3 illustrates the nominal composition of the synthesized MBGNs.

### 4.3. Characterization of Ag-Sr-Doped MBGNs

#### 4.3.1. Morphological Characterization

The surface morphology of the as-prepared nanoparticles as well as those obtained after bioactivity tests was investigated using scanning electron microscopy (SEM; LEO 435VP, Carl Zeiss™ AG, Jena, Germany). First, to make the samples conductive and reduce the effect of charging, the samples were coated with a thin layer (around 10 nm) of gold via the sputtering technique (Q150/S, Quorum Technologies™, Lewes, UK). The SEM images were taken at different magnifications.

To further investigate the morphology of MBGNs, the BET (Brunauer−Emmett−Teller) analysis was carried out. Nitrogen adsorption/desorption was used to measure the pore volume (porosity).

#### 4.3.2. Compositional Analysis

Energy-dispersive X-ray spectroscopy (EDX; X-MaxN Oxford Instruments, Abingdon, UK) was used to determine the composition of the as-synthesized particles. Furthermore, the ratio of the different elements in the hybrid nanoparticles was evaluated. In order to conduct EDX analysis, the synthesized Ag-Sr MBGNs (10 mg) were pressed into the pellets and then EDX analysis was conducted by using the working distance of 6 mm and energy of 25 KV. In order to conduct the EDX analysis after immersion in SBF, the samples were coated with a thin layer (around 10 nm) of gold via the sputtering technique (Q150/S, Quorum Technologies™, Lewes, UK).

Fourier transform infrared spectroscopy (FTIR) measurements were carried out on pellets of Ag-Sr-, Ag-, and Sr-doped MBGNs using the potassium bromide (KBr) disk method on a Shimadzu IRAffinity-1S (Shimadzu Corp, Kyoto, Japan), equipped with Lab Solution IR software and a Quest ATR GS10801-B single-bounce diamond accessory (Specac Ltd. London, UK) at room temperature. In order to prepare samples for FTIR studies, the MBGNs were grounded to a fine powder, followed by mixing with KBr powder in the MBGNs:KBr ratio of 1:100. The mixture was subjected to further grinding to achieve a homogeneous mixture, followed by pressing using a hydraulic pressure of 5 tons/cm^2^ to form the disk samples. The IR transmission spectra were recorded immediately after preparing the discs. For optimal results, the device was cleaned with ethanol before the sample was applied. Furthermore, a background scan was conducted with 128 runs. Every sample was measured with 128 transmittance scans with a resolution of 4 cm^−1^ in Happ-Genzel apodization using wavelengths from 400 to 4000 cm^−1^. To reduce the signal noise, the spectra were smoothed by 15 points.

In addition to FTIR, the samples of Ag-Sr-doped MBGNs were tested with X-ray diffraction (XRD) (MiniFlex 600, Rigaku Corporation, Tokyo, Japan) to characterize the doped MBGNs and pure MBGNs. The diffraction pattern was recorded using Ni-filtered Cu K_α_ radiation (λ = 1.54 Å) operated at 40 kV and 40 mA over the 2θ angular range of 20–80° (with 0.02° step and a speed of 2° per minute).

#### 4.3.3. Zeta Potential

To measure the zeta potential, the Zetasizer Nano zsp (Malvern Panalytical, London, UK) was used. The analyzed suspensions (powder samples in absolute ethanol) were diluted to the ratio of 0.1 g.L^−1^ in particles. Three measurements per suspension at standard pH were taken at a maximum of 100 runs each and averaged. After each measurement, the cell was flushed out with ethanol again.

#### 4.3.4. Ion-Release Profile

To investigate the Ag and Sr ion release from prepared Ag-Sr MBGNs, 75 mg of powder sample was dispersed in SBF solution (50 mL) for different time intervals (day 1, day 3, day 7, day 14, and day 21) in an orbital shaking incubator at 37 °C. Ag and Sr ions released in the medium under dynamic conditions were measured, using ICP-OES (inductively coupled plasma/optical emission spectrometry) from IRIS Advantage, Thermo Jarrell Ash. Approximately 1 g of the sample was dissolved in a 5% HNO_3_ solution and heated gently to ensure complete dissolution. The solution was made up to 50 mL volumetrically and analyzed by ICP-OES (IRIS Advantage, Thermo Jarrell Ash, Waltham, US) against a calibration traceable under ISO: 17,025 guidance.

#### 4.3.5. Antibacterial Studies

To test the antimicrobial properties of synthesized particles, an antibacterial test (called turbidity test) was conducted [48,49]. To grow *E. coli* and *S. Carnasous* bacteria in test tubes, a sterile wooden tip was used to scratch bacteria from a frozen sample and dropped into Lysogeny broth (LB-medium). After an incubation of 24 h, the medium had a high concentration of bacteria and was ready to use. Ag-Sr MBGNs, Ag MBGNs, Sr MBGNs, and Ag-Sr MBGNs were added in the test tubes and incubated for 24 h. Then, an optical density (OD) measurement was performed with absorbance at 600 nm (OD_600_). For each measurement, OD_600_ medium was taken as a reference.

#### 4.3.6. Disc Diffusion Test

Petri dishes were spread homogenously with heated agar inoculated with bacteria (*E. coli* and *S. Carnasous*). Afterwards, the prepared pellets of different concentrations of MBGNs were placed and then incubated at 37 °C for 24 h. After 24 h of incubation, the petri dishes were taken out and digital images were taken to track the zone of incubation.

#### 4.3.7. In Vitro Bioactivity Test

In vitro bioactivity of the synthesized Ag-Sr MBGNs was investigated following Kokubo et al. [50]. The composition of SBF was adopted from Reference [33] and the pH was set at 7.4. The synthesized Ag-Sr MBGNs were pressed into the pellets by using an electrohydraulic pressing device (Mauthe Maschinenbau, Salem, Germany). The prepared pellets were immersed in SBF, and the volume of SBF was set to 1 mg/mL. The SBF solution was changed every three days to simulate a refreshing system. The different sets of samples were taken after 1, 7, and 30 days. The samples were gently washed with (De-ionized) DI-water to prevent salt crystals on the surface and put into the heating stove at 60 °C for drying. After drying, the samples were weighed for the degradation studies and characterized using SEM and EDX analyses.

## Figures and Tables

**Figure 1 gels-07-00034-f001:**
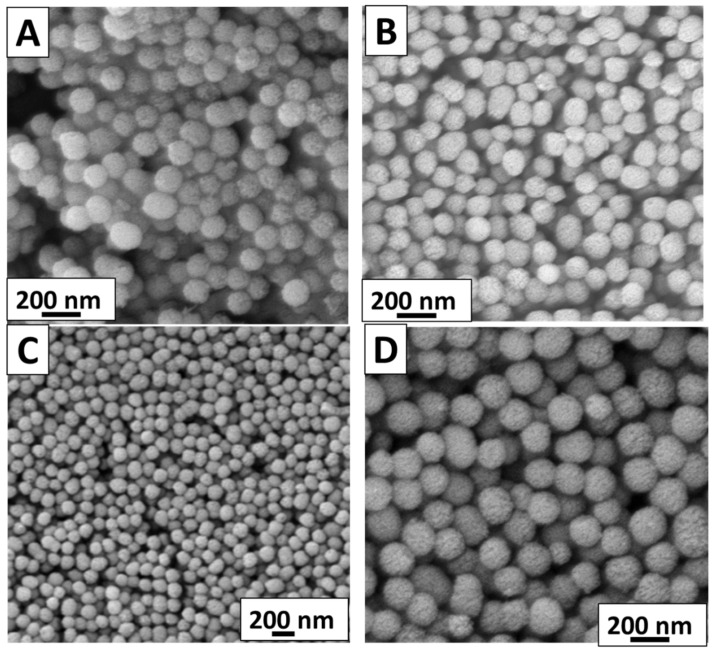
Scanning electron microscopy (SEM) images showing the morphology of the produced nanoparticles: (**A**) pure MBGNs, (**B**) Ag-MBGNs, (**C**) Sr-MBGNs, (**D**) Ag-Sr MBGNs.

**Figure 2 gels-07-00034-f002:**
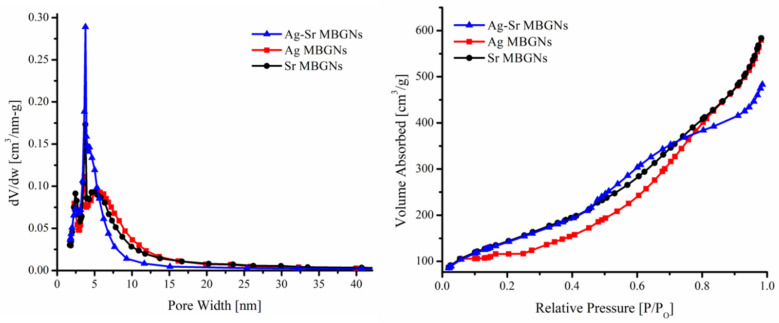
BET (Brunauer-Emmett-Teller) results showing the pore size distribution (**left**) and nitrogen adsorption-desorption isotherms of synthesized bioactive glass nanoparticles (**right**).

**Figure 3 gels-07-00034-f003:**
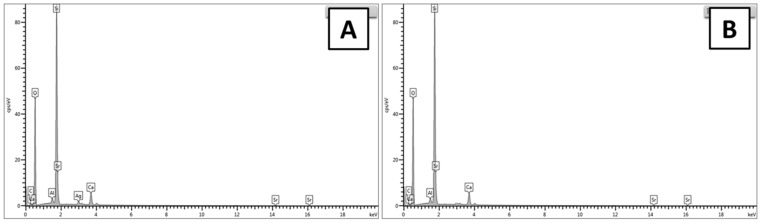
Energy dispersive X-ray (EDX) analysis of the synthesized particles: (**A**) Ag-Sr MBGNs and (**B**) MBGNs.

**Figure 4 gels-07-00034-f004:**
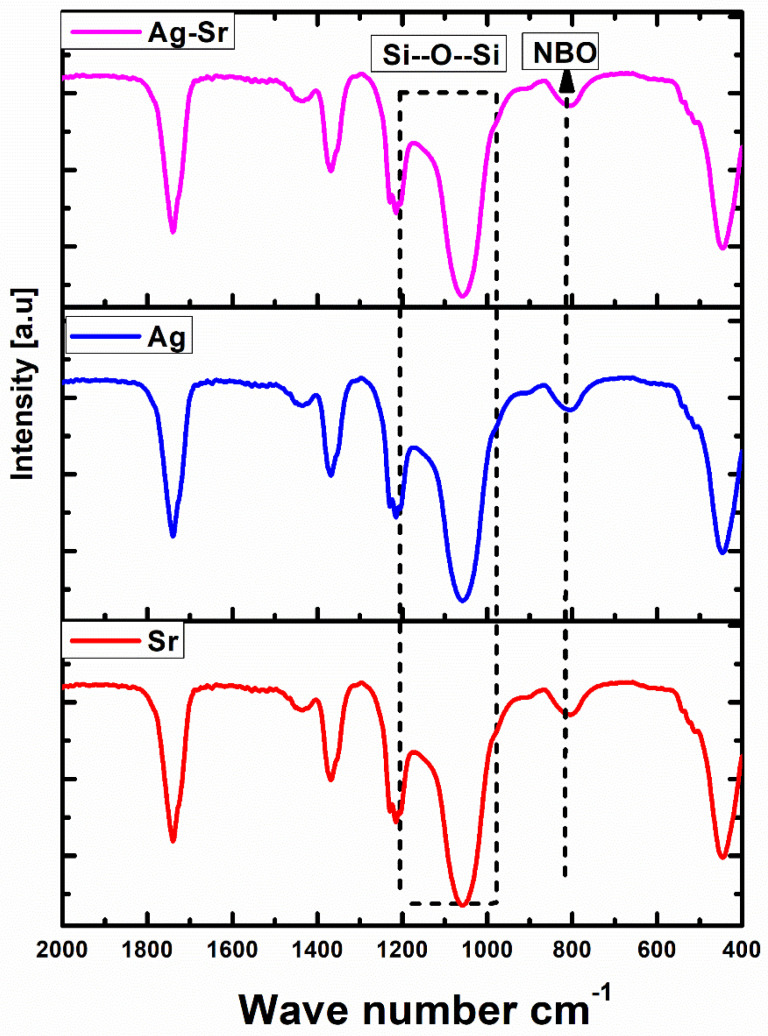
Fourier transform infrared (FTIR) spectroscopy of Ag-Sr, Ag MBGNs, and Sr MBGNs.

**Figure 5 gels-07-00034-f005:**
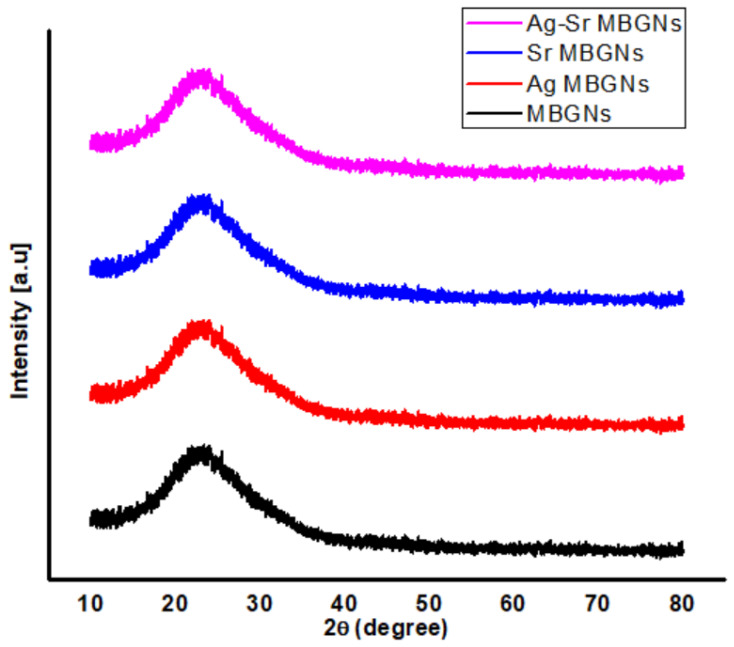
X-ray diffraction (XRD) patterns of Ag-Sr-, Ag-, and Sr-doped MBGNs.

**Figure 6 gels-07-00034-f006:**
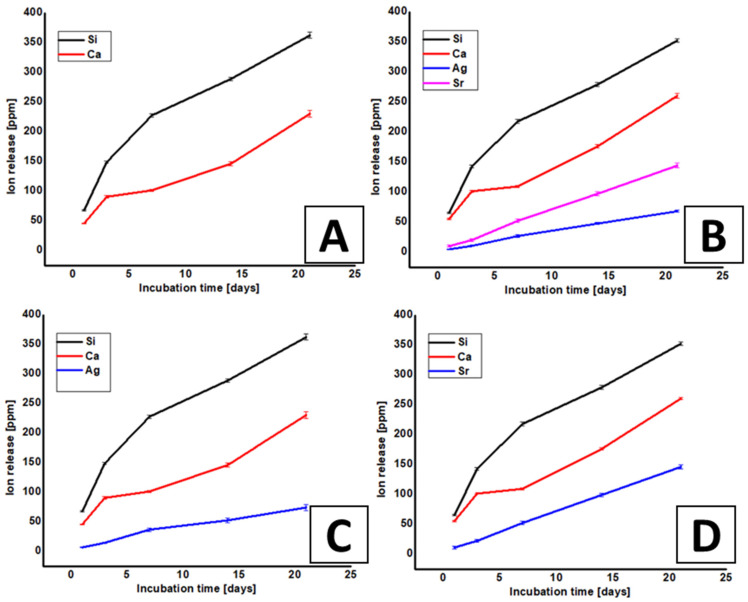
Absolute ion-release profile of Si, Ca, Sr, and Ag ions from (**A**) MBGMNs, (**B**) Ag-Sr MBGNs, (**C**) Ag MBGNs, and (**D**) Sr MBGNs samples immersed in SBF (Simulated Body Fluid) measured by using ICP (Inductively Coupled Plasma) shown in A to D (each experiment was repeated 5 times and the mean values were reported with the standard deviation represented by the error bars in the figure).

**Figure 7 gels-07-00034-f007:**
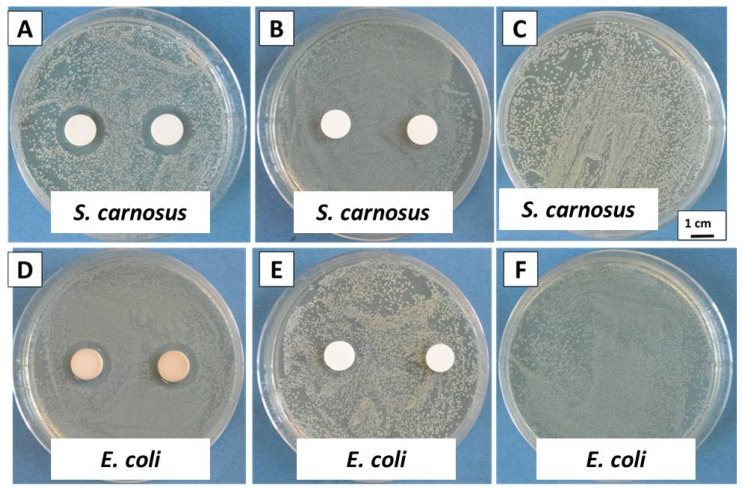
Inhibition halo tests with *S. carnosus* for (**A**) Ag-Sr MBGNs, (**B**) MBGNs, (**C**) reference sample and with *E. coli* for (**D**) Ag-Sr MBGNs, (**E**) MBGNs, (**F**) reference sample.

**Figure 8 gels-07-00034-f008:**
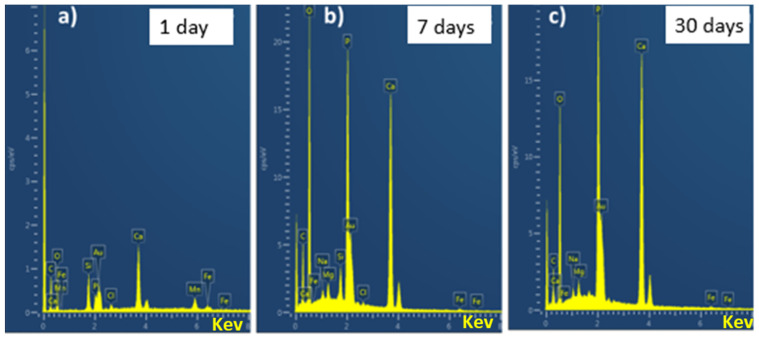
EDX analysis of synthesized Ag-Sr MBGNs after immersion in SBF for (**a**) 1 day, (**b**) 3 days, and (**c**) 1 month.

**Figure 9 gels-07-00034-f009:**
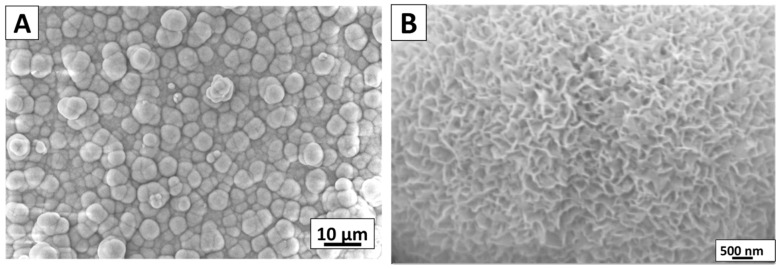
SEM images of Ag-Sr MBGNs pellets after immersion in SBF for 7 days, (**A**) low-magnification image, (**B**) higher magnification image.

**Figure 10 gels-07-00034-f010:**
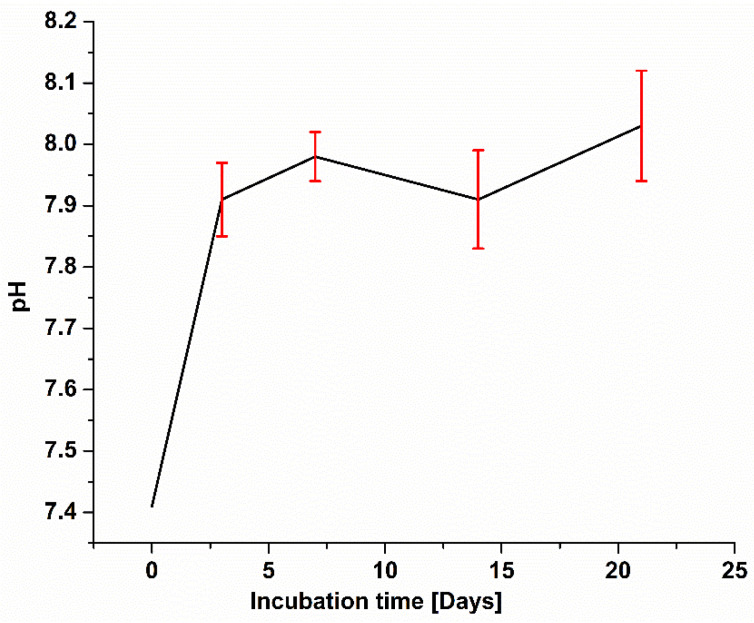
The difference in pH value of SBF after immersion of Ag-Sr MBGNs. pH measured at different time points from 0 to 21 days (the error bar indicates mean ± standard deviation for three individual experiments).

**Figure 11 gels-07-00034-f011:**
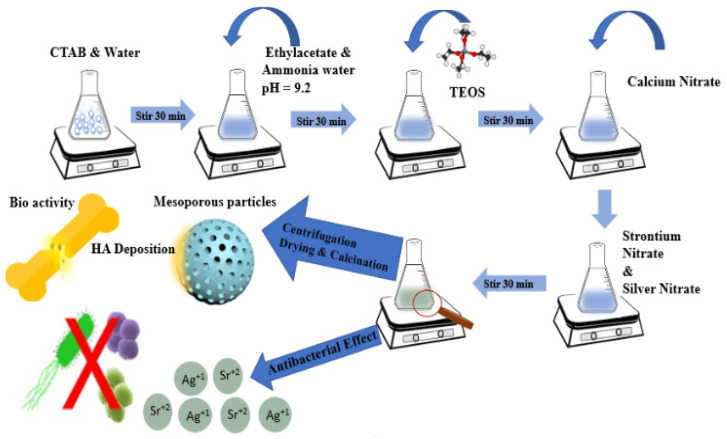
Synthesis of Ag-Sr-doped mesoporous bioactive glass nanoparticles (Ag-Sr MBGNs) modified by the Stöber method.

**Table 1 gels-07-00034-t001:** Zeta potential measurements for MBGNs, Ag MBGNs, Sr MBGNs, and Ag-Sr MBGNs nanoparticles in pure ethanol.

Particles	Zeta Potential ± SD (Standard Deviation) (mV)
MBGNs	22±3
Ag MBGNs	15±2
Sr MBGNs	34±3
Ag-Sr MBGNs	17±2

**Table 2 gels-07-00034-t002:** Results of the turbidity test at optical density of 600 nm (OD_600_) carried out on MBGNs, Ag-Sr MBGNs, Ag MBGNs, and Sr MBGNs after 1, 2, 3, 4, 6, and 24 h of incubation (each experiment is repeated thrice, and the mean value is reported along with the standard deviation).

Time (h)	MBGNs	Ag-Sr MBGNs	Ag MBGNs	Sr MBGNs
1	0.010 ± 0.002	0.020 ± 0.003	0.025 ± 0.004	0.010 ± 0.005
2	0.010 ± 0.003	0.030 ± 0.005	0.025 ± 0.008	0.020 ± 0.007
3	0.040 ± 0.005	0.040 ± 0.005	0.030 ± 0.007	0.055 ± 0.009
4	0.050 ± 0.007	0.010 ± 0.002	0.008 ± 0.003	0.065 ± 0.012
6	0.070 ± 0.008	0.005 ± 0.001	0.003 ± 0.001	0.090 ± 0.032
24	0.190 ± 0.010	0 ± 0	0 ± 0	0.205 ± 0.12

**Table 3 gels-07-00034-t003:** Nominal composition of as-synthesized MBGNs.

Mesoporous Bioactive Glass Nanoparticles Type	Composition (mol.%)			
	SiO_2_	CaO	SrO	AgO
MBGNs	70	30	0	0
5Sr-MBGNs	70	25	5	0
1Ag-MBGNs	70	29	0	1
5Sr-1Ag MBGNs	70	24	5	1

## Data Availability

The additional data relevant to the article is available on request to the corresponding author.

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
