# Peer review of "Synthesis and Characterization of Silver–Strontium (Ag-Sr)-Doped Mesoporous Bioactive Glass Nanoparticles"

_gels, 2021, doi:10.3390/gels7020034_

Round 1
Reviewer 1 Report
Although the work is of potential interest to the journal reasers, the manuscript requires several improvements/clarifications. Specifically: Introduction is too broad and misleading – it should focus on the aim of the work; Description of the ion release profile is unclear and P ions release profile upon SBF incubation could be added; same with FTIR analyses – please improve the method description adding all necessary details. Textural characterization lacks description of BET analysis and Fig. 3 content does not correspond to the text/description. Zeta potential – table – please explain “HA”. The explanation of the zeta potential results is controversial – the difference of 0.05 in electronegativity/positivity should not significantly affect the surface charge. EDX figures are unclear – how did you prepare the samples for EDX analyses? pH measurements – why the pH at point 0 is equal 8.4? What was SBF pH and composition? Again, the figure content (pH) does not match the description (5-10-15 days vs. 3-7-14-21 days). TEM analyses would be also of benefit to assess the porous structure/hierarchy of nanoparticles).
Author Response
ANSWERS TO COMMENTS OF REFEREES
Reviewer #1:
Comments:
Although the work is of potential interest to the journal reasers, the manuscript requires several improvements/clarifications.
- Specifically: Introduction is too broad and misleading – it should focus on the aim of the work;
Our Answer: We have revised the introduction as suggested by the reviewers. Please see the section1 (page 1 - 3)
- Description of the ion release profile is unclear and P ions release profile upon SBF incubation could be added; same with FTIR analyses
Our Answer: The P ion release was not studied because the MBGNs did not contain P, as illustrated in Table 1 of the revised manuscript. FTIR results of Ag-Sr MBGNs incorporated in chitosan/gelatin matrix after immersion in SBF was illustrated in our previous studies. It was observed that the Ag-Sr MBGNs develop HA crystal upon immersion in SBF [1]. FTIR results confirmed the presence of phosphate and carbonate groups upon immersion in SBF. We have revised the manuscript accordingly, please see the section 3.7 (Page 14).
- Please improve the method description adding all necessary details.
Our Answer: We have improved the method description as suggested by the reviewer, please see the section 2.2 (page 3)
- Textural characterization lacks description of BET analysis and Fig. 3 content does not correspond to the text/description.
Our Answer: We have revised the description of BET analysis. Please see the section 3.1 (page7)
- Zeta potential – table – please explain “HA”. The explanation of the zeta potential results is controversial – the difference of 0.05 in electronegativity/positivity should not significantly affect the surface charge.
Our Answer: We thank the reviewer for pointing out the mistake. (‘HA’ was typo error). The word ‘HA’ is replaced with MBGNs in the revised manuscript. Please see the Table 2 (page#10) of the revised manuscript. We have also changed the description on the variation of zeta potential upon addition of Sr. Please see the section 3.3 (page#10) in the revised manuscript.
- EDX figures are unclear – how did you prepare the samples for EDX analyses?
Our Answer: We have added the description for the preparation of samples for EDX analysis. Please the section 2.3.2 (page5).
- pH measurements – why the pH at point 0 is equal 8.4?
Our Answer: We apologize for the mistake we have corrected Figure 11(page 15) in the revised manuscript.
- What was SBF pH and composition?
Our Answer: The pH of SBF was 7.4 and composition is illustrated in Table 1 [reference 33 of the manuscript]. We have revised the manuscript accordingly, please see section 2.3.6 (page 6)
Table 1. Ionic Concentration of SBF
|
Ionic concentration |
(mmol/L) |
|
Na+ |
142.0 |
|
K+ |
5.0 |
|
Ca+2 |
2.5 |
|
Mg+2 |
1.5 |
|
HCO3- |
4.2 |
|
Cl- |
147.8 |
|
HPO4 -2 |
1.0 |
|
SO4 -2 |
0.5 |
- Again, the figure content (pH) does not match the description (5-10-15 days vs. 3-7-14-21 days).
We apologize for the mistake the Figure 11 is revised accordingly, please see the page#15 of the revised manuscript. Please See the section 3.2 (page 16)
- TEM analyses would be also of benefit to assess the porous structure/hierarchy of nanoparticles).
Our Answer: Due to COVID we were not able to TEM analysis, as most of the labs were closed.

Reviewer 2 Report
This paper reports a systematic study of Ag and Sr doped bioactive glasses made using a low temperature modified Stöber process. The results seem to be consistent with previous studies showing the usefulness of silver doping, and of strontium doping of bioactive glasses.
There are a couple of points that need clarification before publication should be considered.
Firstly, how much of the final bioactive glass products were actually made? The amounts of only some of the reactants were described in Section 2.2; the amounts of calcium nitrate, silver nitrate, and strontium nitrate are not stated. 3 ml of TEOS does not seem sufficient to provide adequate samples for the subsequent characterization techniques used in the study.
Secondly, one might quibble about concluding that “… Ag and Sr was released in a sustained manned thus, providing long-term therapeutic effect” when the longest time investigated in the study was only 24 hours.
Thirdly, the English could be improved. For example, the sentence in the first paragraph of the Introduction: “Moreover, the percentage of people over 50 years are suffering from diseases like osteoporosis, osteoarthritis, osteomalacia, bone cancer and other musculoskeletal diseases [3].” What is it about the percentage? Increasing, decreasing? Small? Large?
What does “livable" mean in: ”…the implant material must be livable to form bone-forming tissues, …”
Author Response
Reviewer #2:
This paper reports a systematic study of Ag and Sr doped bioactive glasses made using a low temperature modified Stöber process. The results seem to be consistent with previous studies showing the usefulness of silver doping, and of strontium doping of bioactive glasses.
There are a couple of points that need clarification before publication should be considered.
- Firstly, how much of the final bioactive glass products were actually made? The amounts of only some of the reactants were described in Section 2.2; the amounts of calcium nitrate, silver nitrate, and strontium nitrate are not stated. 3 ml of TEOS does not seem sufficient to provide adequate samples for the subsequent characterization techniques used in the study.
Our Answer: In the one run 1-1.5 grams of bioactive glass powder was produced. However, we have run large number of cycles (20 runs at each composition) to prepare the bioactive glass powder for this study. We have revised this section as suggested by the reviewer. Please see the section 2.2 (page3).
- Secondly, one might quibble about concluding that “… Ag and Sr was released in a sustained manned thus, providing long-term therapeutic effect” when the longest time investigated in the study was only 24 hours.
Our Answer: We are interested in bioactivity and wants to eradicate biofilm so initial burst release is necessary for preventing the biofilm formation. Furthermore, the ion release study was conducted up to 21 days which led to the conclusion that Ag and Sr was release in sustained manner. The amount of silver released after 21 days of incubation was still above the minimum inhibitory concentration, which refers to that the material will be antibacterial. The manuscript is revised accordingly, please see section 3.4 (page12)
- Thirdly, the English could be improved. For example, the sentence in the first paragraph of the Introduction: “Moreover, the percentage of people over 50 years are suffering from diseases like osteoporosis, osteoarthritis, osteomalacia, bone cancer and other musculoskeletal diseases [3].” What is it about the percentage? Increasing, decreasing? Small? Large?
Our Answer: We have revised the introduction completely as suggested by the reviewer 1 also. The English of the whole manuscript was checked carefully.
- What does “livable" mean in: ”…the implant material must be livable to form bone-forming tissues, …”
Our Answer: We have revised the introduction section completely. Please see the revised introduction section.
Round 2
Reviewer 1 Report
Although the revised version has been improved, the manuscript still requires some additional info as listed below:
FTIR analysis – substantial errors still detected:" The bands around 455 cm-1 and 1067 cm-1 can be assigned to Si–O–Si stretching and Si–O–Si rocking
modes, respectively [36]" - This is mistaken: 455 rocking, 1067 stretching. "The broad band at 1200 to 1000 cm-1 depicts Si-O-Si vibrations [37]. Intensity of non-bridging oxygen (NBO) peak observed at 799 cm-1 for MBGNs in silica network" - The peak around 800cm-1 is assigned to the Si–O–Si bridging bonds in the SiO4 tetrahedrons not to NBO
The Authors continue to write about "high specific surface area" but there are no values regarding that provided; “The specific surface area of Sr, Ag, and Ag-Sr doped MBGNs was calculated via the Brunauer-Emmett-Teller (BET) method using the nitrogen desorption branch of the isotherm” – where do the authours present the calculation of SSA???
Fig. 3 has mistakes in the figure captions again
ICP of SBF - P should be analysed because the consumption of phosphate ions from SBF confirmes the HA formation; additionally, please explain why the Si concentration in SBF for starting time point is higher than Ca (it follows from the presented results – poor quality of figure, without labelling) whereas SBF does not contain Si ions in any form????
EDX – what kind of coatings were used to perform SEM/EDX analyses? Result presented in Fig.4 suggest a carbon layer, Fig. 9 – is this gold??? Please add appropriate descriptions to the methods section
Fig. 4/9 EDX analyses lack of x-axis label and units, poor figures quality - not readable, lack of peaks description, please revise and supplement
Author Response
Referee comments
Reviewer 1
- FTIR analysis – substantial errors still detected:" The bands around 455 cm-1 and 1067 cm-1 can be assigned to Si–O–Si stretching and Si–O–Si rocking modes, respectively [36]" - This is mistaken: 455 rockings, 1067 stretching. "The broad band at 1200 to 1000 cm-1 depicts Si-O-Si vibrations [37]. Intensity of non-bridging oxygen (NBO) peak observed at 799 cm-1 for MBGNs in silica network" - The peak around 800cm-1 is assigned to the Si–O–Si bridging bonds in the SiO4 tetrahedrons, not to NBO.
Our Answer: We are really thankful to the reviewer for raising an important point. We have introduced the corrections please see Section 3.2 (page#8).
- The Authors continue to write about "high specific surface area" but there are no values regarding that provided; “The specific surface area of Sr, Ag, and Ag-Sr doped MBGNs was calculated via the Brunauer-Emmett-Teller (BET) method using the nitrogen desorption branch of the isotherm” – where do the authors present the calculation of SSA???
Our Answer: We apologize for the mistake. The BET result/method section is revised accordingly. Please see the Section 3.1 (Page#7).
- Fig. 3 has mistakes in the figure captions again.
Our Answer: We apologize for the mistake. The caption of Figure 3 is revised please see the revised manuscript (page#8).
- ICP of SBF - P should be analysed because the consumption of phosphate ions from SBF confirmes the HA formation; additionally, please explain why the Si concentration in SBF for starting time point is higher than Ca (it follows from the presented results – poor quality of the figure, without labeling) whereas SBF does not contain Si ions in any form????
Our Answer: ICP analysis for P release was not carried out in the current study, as we initially studied the release of ions from MBGNs and not the interaction of ions from SBF to MBGNs. It would be interesting in the future to carry out the suggested study. But currently, due to the third wave of COVID in Pakistan labs are closed and no more experiments are possible. We have revised the manuscript please see section 3.4 (page#12).
I think here is the confusion; we investigate the release of ions from MBGNs to SBF and not the ion release from SBF to MBGNs. Yes, SBF does not contain Si ions but MBGNs contain 70% silica and MBGNs tend to degrade rapidly owe to their morphological features. Due to the rapid degradation of MBGNs, the Si ions were released from MBGNs to SBF at a rapid pace. Furthermore, MBGNs contain a higher amount of silica compared to Calcium. Thus the release of Ca ions was lower compared to the Si ions, which is in agreement with the previous studies (ref.27 & 46 of the manuscript).
- EDX – what kind of coatings were used to perform SEM/EDX analyses? Result presented in Fig.4 suggests a carbon layer, Fig. 9 – is this gold??? Please add appropriate descriptions to the methods section.
Our Answer: We have revised Figure 4 in the revised manuscript (page#8). EDX analysis was carried out without sputtering for Figure 4. However, after incubation in SBF, the charging effect dominates, and thus it was necessary to sputter the samples with the gold. The manuscript is revised accordingly. Please see section 2.2 (page#5).
- Fig. 4/9 EDX analyses lack x-axis label and units, poor figures quality - not readable, lack of peaks description, please revise and supplement
Our Answer: Here the EDX analysis was only used as a qualitative tool for the detection of elements. The quantitative EDX analysis was not carried out in the current study. Therefore, we focused on presenting the peaks of the relevant elements. We have revised Figure 4 and Figure 7 with peaks labeled and X-axis unit’s visible (page# 8 & 14, respectively)